# Rust Fungi on Medicinal Plants in Guizhou Province with Descriptions of Three New Species

**DOI:** 10.3390/jof9090953

**Published:** 2023-09-21

**Authors:** Qianzhen Wu, Minghui He, Tiezhi Liu, Hongmin Hu, Lili Liu, Peng Zhao, Qirui Li

**Affiliations:** 1State Key Laboratory of Functions and Applications of Medicinal Plants, Guizhou Medical University, Guiyang 550004, China; wqz1665@aliyun.com (Q.W.); minghuihegmu1979@163.com (M.H.); a2942338310@aliyun.com (H.H.); lililiu550025@163.com (L.L.); 2The High Efficacy Application of Natural Medicinal Resources Engineering Center of Guizhou Province (The Key Laboratory of Optimal Utilization of Natural Medicine Resources), School of Pharmaceutical Sciences, Guizhou Medical University, University Town, Guian New District, Guizhou 550004, China; 3College of Chemistry and Life Sciences, Chifeng University, Chifeng 024000, China; tiezhiliu@aliyun.com; 4Immune Cells and Antibody Engineering Research Center of Guizhou Province, Guizhou Medical University, Guiyang 550004, China; 5Key Laboratory of Biology and Medical Engineering, Guizhou Medical University, Guiyang 550004, China; 6State Key Laboratory of Mycology, Institute of Microbiology, Chinese Academy of Sciences (CAS), Beijing 100101, China

**Keywords:** medicinal plant, molecular phylogeny, new taxa, phytopathogen, *Pucciniales*

## Abstract

During the research on rust fungi in medicinal plants of Guizhou Province, China, a total of 9 rust fungal species were introduced, including 3 new species (*Hamaspora rubi-alceifolii*, *Nyssopsora altissima*, and *Phragmidium cymosum*), as well as 6 known species (*Melampsora laricis-populina*, *Melampsoridium carpini*, *Neophysopella ampelopsidis*, *Nyssopsora koelrezidis, P. rosae-roxburghii*, *P. tormentillae*). Notably, *N. ampelopsidis* and *P. tormentillae* were discovered for the first time in China, while *M. laricis-populina*, *Me. carpini*, and *Ny. koelreuteriae* were first documented in Guizhou Province. Morphological observation and molecular phylogenetic analyses of these species with similar taxa were compared to confirm their taxonomic identities, and taxonomic descriptions, illustrations and host species of those rust fungi on medicinal plant are provided.

## 1. Introduction

Rust fungi comprise the largest and most ubiquitous group of obligately biotrophic fungi on vascular plants [1]. The economic impact of rust fungi cannot be ignored. Because of rust fungi, many economic plants suffer diseases and reduce yields [2]. For example, when wheat is harmed by *Puccinia striiformis* Westend., its yield can be reduced up to 50% [3]. The impact of rust fungi on tropical crops is also immeasurable [4]. The coffee rust fungus has a global distribution and is often found in the coffee growing areas of China, with a greater impact on the main coffee producing countries [2,5,6].

More than 8000 species of rust fungi have been identified, mainly on the basis of their morphological characteristics of teliospores and spermonogia, however, there are still a larger number of genera *incertae sedis* [7,8,9,10,11]. Rust occurs on ferns to advanced monocots and dicots, and they are obligate biotrophic phytopathogens that produce not only basidiospore but also four other different types of spores: aeciospore, urediniospore, teliospore, and spermatia [8,12,13]. Many species in the order *Pucciniales* were not described with all types of spores, and have various lifestyles (micro-, hemi-, demi-, or macrocyclic), with alternation on single (autoecious) or two unrelated host plants (heteroecious) [7,14]. To date, approximately 8400 rust species are currently recognized worldwide, and 71 genera and 1175 species have been discovered in China so far [11]. However, species diversity, host alternation and geographic distribution of rust fungi in China remain poorly understood. 

Medicinal plants are also infected by rust fungi on a large scale [15]. By the end of 2020, a total of 79 rust species have been reported on 76 medicinal plant species from 33 families, and these rust fungi restricts the development and utilization of medicinal plants and affects the quality of botanicals [16,17]. In recent years, as many as 3924 species of medicinal plants are cultivated in Guizhou province, and the number of medicinal plants is increasing number year by year [18]. As important pathogenic fungi, rust can infect the leaf and stem of a variety of medicinal plants and affect their quality and yield which eventually hampered the development and utilization of medicinal plant resources [19]. For example, rust diseases are frequently found on pepper leaves in Guizhou province, and those diseases seriously affect the normal development of pepper, with their incidence reaching 90% in serious infections [20]. However, there are few researches on rust species infecting medicinal plant diseases in Guizhou province [21]. Therefore, it is of great significance to investigate the species diversity of rust fungi on important medicinal plants for local medicinal production.

In 2021, an investigation of rust fungi on the medicinal plants was carried out in Guizhou province, China. Nine species including three new species were found on medicinal plants. Detailed descriptions and illustrations of all those novel species and other species on the medicinal plants are provided.

## 2. Materials and Methods

### 2.1. Sample Collections

Rust infected specimens were collected from Guiyang, Qingzhen and Anshun cities in Guizhou province, China. All hosts and habitats information of specimens was recorded. For each specimen, part of specimens was kept in a refrigerator at 4 °C, and the other part was made as a dry specimen [22]. Specimens were deposited in both Mycological Herbarium of the Chifeng University, Inner Mongolia, China (CFSZ) and Herbarium of Guizhou Medical University (GMB).

### 2.2. Morphology

The specimens were observed under a stereomicroscope (Nikon SMZ745T, Nikon Corporation, Tokyo, Japan) and shot with a Canon digital camera (Canon EOS 1500D, Canon Inc., Tokyo, Japan) fitted on. Microscope images of the samples were taken by a Canon EOS 700D digital camera fitted on the Nikon ECLIPSE Ni compound microscope (Nikon, Japan). Measurements were taken with the Tarosoft (R) Image Frame Work (v.0.9.7). More than 30 morphological characteristics such as teliospores, urediniospores, and paraphyses were measured for each specimen. Photo plates were arranged by using Adobe Photoshop CS6 v. 13 (Adobe Systems Software Ireland Ltd, San Jose, USA). The different spore stages of rust fungi are designated by the following Roman numerals: spermogonia/spermatia (0), aecia/aeciospores (I), uredinia/urediniospores (II), telia/teliospores (III), and basidia/basidiospore (IV). We applied the definitions of spore stage based on Cummins and Hiratsuka [7], and followed morphological types of spermogonia designated by Hiratsuka and Hiratsuka [23].

### 2.3. DNA Extraction, Polymerase Chain Reaction (PCR), and Sequencing

The rust sori were picked out into a sterilized centrifuge tube with a sterilized fine needle for DNA extraction. Genomic DNA was extracted following the manufacturer’s protocol of the OMEGA E.Z.N.A.^®^ Fungal Genomic DNA Extraction Kit (D3390, Guangzhou Feiyang Bioengineering Co., Ltd., Guangzhou, China). DNA extracts were stored at –20 °C. PCR was carried out in a volume of 25 μL containing 9.5 μL of ddH_2_O, 12.5 μL of 2 × *Taq* PCR Master Mix (2 × Taq Master Mix with dye, TIANGEN, China), 1 μL of DNA extraction and 1 μL of forward and reverse primers (10 µm each) in each reaction. Primers pairs, ITS4/ITS5 and LR0R/LR5 (Sangon Biotech, Shanghai, China) were used to amplify the regions of internal transcribed spacer (ITS) and large subunit ribosomal (LSU), respectively [8,24,25,26]. PCR profiles for the ITS and LSU were: initially at 95 °C for 5 min, followed by 35 cycles of denaturation at 94 °C for 1 min, annealing at 52 °C for 1 min, polymerization at 72 °C for 1.5 min and a final extension at 72 °C for 10 min. PCR products were sequenced by Sangon Biotech (Shanghai) Co., Ltd., China.

### 2.4. Phylogenetic Studies

All sequences used for phylogenetic tree construction were listed in Table 1. Sequences were aligned by MAFFT v. 7.394 (https://www.ebi.ac.uk/Tools/msa/mafft/) (accessed on 12 September 2023) [27] and adjusted to ensure maximum similarity using TrimAl v1.4.1 [28]. Alignments were converted from FASTA to PHYLIP format by using Alignment Transformation Environment online (https://sing.ei.uvigo.es/ALTER/) (accessed on 12 September 2023) [29]. Maximum Likelihood (ML) analyses and Bayesian posterior probabilities (BYPP) based on a combination of ITS and LSU sequence data were performed using RAxML-HPC 7.4.2 BlackBox [30] and MrBayes v. 3.2.7 tools in the CIPRES Science Gateway platform [31,32]. GTR+I+G was estimated as the best-fit substitution model by jModelTest2 on XSEDE v.2.1.6 [33,34]. The Bootstrap values of ML analyses were obtained by running 1000 replicates by using a Markov chain Monte Carlo (MCMC) method to approximate the posterior probabilities of trees. Six simultaneous Markov Chains were run for 3,000,000 generations and trees were sampled every 1000th generation. Finally, the trees were visualized in FigTree v.1.4.4 [35] and edited by using Adobe Photoshop CS6 v. 13 software. The final alignment and phylogenetic trees were deposited in TreeBASE v. 2 under the submission ID30041 (http://www.treebase.org/) (accessed on 12 September 2023). 

## 3. Results

In this study, 11 samples were collected from 9 species of medicinal plants in nine genera in seven families in Guizhu Province. Ten species including three new species were identified based on morphological and molecular phylogenetic studies. Morphological and phylogenetically allied taxa were selected for the final phylogenetic analyses, mainly following Zhao et al. [11]. The alignment for *Melampsoraceae* includes 1200 character (ITS: 532 bp, LSU: 1200 bp) (Figure 1). The dataset of phylogenetic tree from *Neophysopellaceae* and *Phakopsoraceae* has 1162 characters (ITS: 474 bp, LSU: 1162 bp) including gaps (Figure 2). The alignment for *Neophysopellaceae* and *Phakopsoraceae* phylogenetic tree contained 20 taxa. The taxa from genera *Gerwasia*, *Hamaspora*, *Kuehneola*, *Phragmidium*, *Trachyspora*, and *Xenodochus* were included in the phylogenetic tree of the family *Phragmidiaceae* (Figure 3). The alignment is made up of 71 species and has 1244 characters including gaps (ITS: 392 bp, LSU: 1244 bp). The phylogenetic tree of the families *Pucciniastraceae* and *Hyalopsoraceae* includes the taxa from genera *Coleopuccinia*, *Hyalopsora*, *Melampsorella*, *Melampsoridium*, and *Pucciniastrum* (Figure 4). In the phylogenetic tree of *Pucciniastraceae*, there are 20 taxa with 1244 characters, including gaps (ITS: 351 bp, LSU: 1244 bp). Twenty-seven representative species from the *Gymnosporangiaceae*, *Sphaerophragmiaceae*, and *Uredinineae incertae sedis* were chosen for Figure 5, which included 1272 characters including gaps (ITS: 398, LSU: 1272). Based on morphological and multi-locus phylogenetic characterisation, 13 specimens were identified to 9 species in 6 genera 5 families (*Hamaspora rubi-alceifolii* sp. nov., *Nyssopsora altissima* sp. nov., *Phragmidium cymosum* sp. nov., *Melampsora laricis-populina*, *Melampsoridium carpini*, *Neophysopella ampelopsidis*, *Ny. koelreuteriae*, *P. rosae*-*roxburghii* and *P. tormentillae*). Among them, three novel species were recognized, moreover, 2 species were reported first in China, and 4 species were first reported on medicinal plant.

### Taxonomy

Based on morphology and molecular phylogeny, all collected rust specimens on medicinal plants were identified as three new species (*Hamaspora rubi-alceifolii* sp. nov., *Nyssopsora altissima* sp. nov., *Phragmidium cymosum* sp. nov.) and six known species (*Melampsora laricis-populina*, *Melampsoridium carpini*, *Neophysopella ampelopsidis*, *Ny. koelreuteriae*, *P. rosae*-*roxburghii* and *P. tormentillae*) All species parasitic on medicinal plants are described and illustrated below.

***Hamaspora rubi-alceifolii*** Q. Z. Wu, T. Z. Liu, P. Zhao & Q. R. Li, **sp. nov**. Figure 6

MycoBank number: MB847104

Etymology: Epithet follows the epithet of host species, *Rubus alceifolius* Poir.

Holotype: GMB0109

*Parasitic* on the leave and stem of *R. alceifolius*. *Telia* up to 5 mm long, mostly hypophyllous, occasionally amphigenous and stem, caespitose, filiform, fluffy, golden yellow when fresh, white when dry; *Teliospores* 116–230 × 20–10 µm (av. = 184 × 17 µm, n = 30), fusiform, hyaline 3–6 septate, mostly 5–6 septate, smooth, the contents yellow when fresh, solid apex 9–30 µm (av. = 23 µm, n = 30). Spermogonia, aecia and uredinia were not observed.

Materials examined: CHINA, Guizhou Province, Guiyang City, Campus of Guizhou Medical University (26°22′48.37″ N, 106°37′30.33″ E), III on leaf of *R. alceifolius* Poir., 6 October 2021, Q. Z. Wu and L. L. Liu, GMB0109, holotype, CFSZ 50531, isotype; CHINA, Guizhou Province, Guiyang City, Huaxi District, III on leaf of *R. alceifolius* Poir., 20 October 2021, Q. Z. Wu and L. L. Liu, GMB0116.

Notes: *Hamaspora rubi-alceifolii* is characterized by 5–6 septate teliospore with long solid apex up to 30 µm, and hypophyllous telia (Figure 6). Phylogenetically, it formed a distinct clade sister to *H. acutissima* with high support values (100% ML, 1 BYPP; Figure 3). Morphologically, the differences between *H. rubi-alceifolii* and *H. acutissima* are in the number of septa in the teliospores (5–6 vs. 2–3), smaller teliospores (116–230 × 10–20 µm vs. 158–205 × 18–25 µm), and smaller solid apex (9–30 µm vs. 20–40 µm). Both *H. rubi-sieboldii* and *H. rubi-alceifolii* exhibit similar teliospore morphology with consistent solid apex size. However, the difference between *H. rubi-alceifolii* and *H. rubi-sieboldii* is that the former has more cells (5–6 vs. 4) and smaller solid apex (116–230 × 10–20 µm vs. 118–240 × 15–23 µm) [73,74,75,76].

Roots and leaves of *Rubus alceifolius* is a traditional Chinese medicine, which can be used for treatment of acute and chronic hepatitis, hepatosplenomegaly and other liver damage diseases [77,78,79,80,81]. Previously, thirteen *Hamaspora* species have already been reported on *Rubus* species [76]. For the convenience of recognition, a worldwide identification key for the *Hamaspora* has been provided.

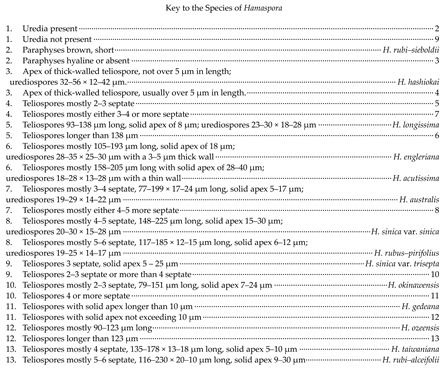


***Melampsora laricis-populina*** Kleb., Z. PflKrankh. 12: 43 (1902) Figure 7

Mycobank number: MB18930

*Uredinia*, mostly hypophyllous, seldom epiphyllous, in little groups, 0.25 mm, Bright yellow. Urediniospores oblong to broadly ellipsoid or obovoid, hyaline to light yellow with yellowish granules, 33–45 × 21–26 µm (av. = 37 × 24 µm, n = 30), wall 1.8–3 µm thick, echinulate except at the smooth apex; Paraphyses clavate to capitate, hyaline to pale yellow, 50–71 × 12–20 µm (av. = 62 × 17 µm, n = 30), wall thick, up to 9 µm at the apex. 

Materials examined: CHINA, Guizhou Province, Kaili City, Xiasi Town (26°26′29.48″ N, 107°47′30.33″ E) II on *Populus lasiocarpa* Oliv., 1 October 2021, Q. Z. Wu, GMB0097.

Notes: *Melampsora laricis-populina* is characterized by the echinulate urediniospores that are obovate or oval, with golden yellow cytoplasm; roughly spherical paraphyses with swollen tips. The uredinial morphologies of our specimen (GMB0097) are consistent with those of *M. laricis-populina* [82,83]. Phylogenetic tree (Figure 1) showed that our specimen (GMB0097) was clustered with *M. laricis-populina* with the high bootstrap supports (83/-). Thus, here we confirmed the rust fungus on *Populus lasiocarpa* as *M. laricis-populina. Populus lasiocarpa* has hemostatic function and mainly used for treatment for bleeding from trauma [84,85]. Previously, *M. laricis-populina* has already been reported on *Populus lasiocarpa* in Japan [86] and Norway [87], and it has been reported in northwest region of China [88].

***Melampsoridium carpini*** (Nees) Dietel, in Engler & Prantl, Nat. Pflanzenfam., Teil. I (Leipzig) 1(1**): 551 (1900) Figure 8

Basionym: *Caeoma carpini* Nees, Syst. Pilze (Würzburg): 16 (1816) [1816-17]

Mycobank number: 205589

*Uredinia*, hypophyllous, scattered or grouped on yellow spots, 0.2 mm diam., yellow. Urediniospores long obovoid, clavate or pear shaped, 19–31 × 13–18 µm (av. = 26 × 15 µm, n = 30), yellow; spore wall hyaline, echinulate, on apex smooth, 0.8–1.5 µm thick. 

Materials examined: CHINA, Guizhou Province, Qingzhen City, Xiasi Town (26°27′18.48″ N, 107°20′7.33″ E) II on *Carpinus turczaninowii* Hance, 8 October 2021, Q. Z. Wu and L. L. Liu, GMB0112, CFSZ 50543.

Notes: *Carpinus turczaninowii* is commonly used to treat bruises, canker sores and swellings, as recorded in the Pharmacopoeia of the People’s Republic of China. *Carpinus turczaninowii* can alleviate arterial damage and inflammation caused by hyperglycemia [89]. It contains *Pheophorbide* A, which has anti-cancer and anti-inflammatory activity [90,91]. We collected rust infected *Carpinus turczaninowii* in Guizhou province in China, and the urediospores of our specimen (GMB0112) are consistent with those of *Me. carpini,* which is characterized by sparse thorns on the surface and a smooth top of urediospores [92,93,94]. According to the phylogenetic tree (Figure 4.), the new collection (GMB0112) was clustered with *Me*. *carpini* with the high bootstrap values (99/-). This species has been found in Anhui, Chongqing, Sichuan, Taiwan and other provinces in China [94], which is the first record of *Me. carpini* from Guizhou province, China.

***Neophysopella ampelopsidis*** (Dietel & P. Syd.) Jing X. Ji & Kakish., in Ji, Li, Li & Kakishima, Mycol. Progr. 18(6): 863 (2019) Figure 9

Basionym: *Phakopsora ampelopsidis* Dietel & P. Syd. [as ’ampelosidis’], in Dietel, Hedwigia 37: 217 (1898)

Mycobank number: MB830298

*Uredinia* hypophyllous, grouped on brown or yellow spots. the paraphyses, incurved, 30–43 × 6–14 µm (av. = 37 × 11 µm, n = 30), wall 2–3 µm thick (av. = 2.5 µm, n = 30). *Urediniospores*, obovoid or obovoid-ellipsoid, dark yellow or brown, 20–32 × 15–21 µm (av. = 25 × 18 µm, n = 30), walls hyaline, echinulate. The wall was colorless or pale yellowish, equally ca 0.8–1.5 µm thick, and evenly echinulate.

Materials examined: CHINA, Guizhou, Qingzhen City, Xiasi Town (26°27′18.18″ N, 107°20′7.13″ E) II on *Ampelopsis sinica* (Mig.) W.T. Wang., 22 July 2021. Q. Z. Wu and L. L. Liu, GMB0110, CFSZ 50532.

Notes: *Ampelopsis sinica* root (ASR) is a traditional Chinese medicine known to have a hepatoprotective function. Moreover, it has been proven having anti-hepatocellular carcinogenic activity and to inhibit Hepatitis B virus activity [95,96,97]. Our rust collection (GMB00110) on *Ampelopsis sinica* is compatible with *Neophysopella ampelopsidis* [98]. Phylogenetically, our collection clustered with *N. ampelopsidis* (IBA-8618) with the high bootstrap supports (100/1, in Figure 2). *N. ampelopsidis* has been previously introduced from Japan, Philippines and Taiwan provinces of China [98]. This is the first record of *N. ampelopsidis* from the Chinese mainland.

***Nyssopsora altissima*** Q. Z. Wu, T. Z. Liu, P, Zhao & Q. R. Li, **sp. nov.**
Figure 10

MycoBank number: MB847103 

Etymology: Epithet follows the epithet of host species, *Ailanthus altissima* (Mill.) Swingle. 

Holotype: GMB0103

*Parasitic* on the leave of *A. altissima*. *Uredinia*, usually amphigenous, sparse or aggregated, pulverulent, golden; *Urediniospores* subglobose or ellipsoid, 18–23 × 16–24 µm (av. = 21 × 20 µm, n =30), cell wall 1.2–2.4 µm (av. = 1.5 µm, n = 30) thick, echinulate. *Telia* soft hypophyllous, aggregated, rounded, pulverulent, dark; *Teliospores* subglobose or globose-trigonal, septa constricted slightly, 2–3 cells are mostly 3, reddish-brown to opaque, 28–38 × 20–39.5 µm (av. = 33.5 × 33 µm, n = 30), projections up to 14 (av. = 12, n = 30), 1–6 (av. = 2, n = 30) apical furcations, 2.8–7.8 µm (av. = 4.5 µm n = 30) long and 1.3–3.3 µm (av. = 2 µm, n = 30) thick, each cell has 1–2 germination pores; pedicel hyaline, persistent, about 33–57 µm (av. = 44 µm, n = 30) long, 5.5–9 µm (av. = 7.8 µm, n = 30) thick.

Materials examined: CHINA, Guizhou Province, Guiyang City, Campus of Guizhou Medical University (26°22′48.37″ N, 106°37′30.33″ E), II, III on leave of *A. altissima*, 8 July 2021, Q.Z. Wu and L.L. Liu, GMB0103, holotype, CFSZ 50535, isotype.

Notes: Phylogenetically, *Nyssopsora altissima* was phylogenetically allied to *Ny. echinata* with a high bootstrap support (86/0.99, Figure 5). Morphologically, *Ny. altissima* differs from *Ny. echinata* by the number of germination pores in teliospores (3 vs. 2), the bigger teliospores (28–38 × 20–39.5 µm vs. 25–30 × 23–27 µm), and the shorter processes (2.8–7.8 µm vs. 6.6–12 µm). Morphologically, *Ny. altissima* can be distinguished from *Ny. cedrelae* by several morphological differences that bigger urediniospores (18–23 × 16–24 µm compared to 15–19 × 13–17 µm), smaller teliospores (28–38 × 20–39.5 µm vs. 28–45 × 18–43 µm), and fewer projections (1–14 vs. 15–22). Additionally, *Ny. altissima* has shorter hyaline (33–57 µm vs. 40–65 µm) and finer hyaline (5.5–9 µm vs. 10–12 µm) than those of *Ny. cedrelae* [76,99].

*Ailanthus altissima* has been used as a medicinal herb to hemostasis and Anti-diarrhea documented in Chinese pharmacopoeia [100]. *Ailanthus altissima* has the anti-malarial, anti-viral and anti-tumor active ingredient and shows potential as a novel drug for the treatment of prostate cancer [101,102,103]. To date, another rust species, *Ny. cedrelae*, has already been reported on *Ailanthus altissima* [76], here we reported another *Nyssopsora* species on *Ailanthus altissima.* Compared to *Ny. cedrelae*, *Ny. altissima* exhibits larger-sized urediniospores (18–23 × 16–24 µm vs. 15–19 × 13–17 µm), shorter apical furcations (2.8–7.8 µm vs. 2–10 µm), and a shorter pedicel (33–57 µm vs. 120 µm).

***Nyssopsora koelreuteriae*** (Syd. & P. Syd.) Tranzschel, J. Soc. bot. Russie 8: 132 (1925) [1923] Figure 11

Basionym: *Triphragmium koelreuteriae* Syd. & P. Syd. 1913

Mycobank number: MB335240

*Parasitic* on leaves of *Koelreuteria bipinnata*, surrounded by yellowish margins spermagonia, aecia and uredinia unknown. *Telia* amphigenous, scattered or slightly clustered, densely aggregated in groups, confluent, naked, erumpent, blackish, chestnut-brown. *Teliospores* 27–31 × 24–32 µm (av. = 28 × 27 µm, n = 30), 3-celled with a single proximal cell and two collateral distal cells, triquetrous pyriform, strongly constricted at septa, blackish–brown; walls uniformly 1.5–3 µm thick, pale yellow when young and becoming blackish–brown when older. Projections up to 18 and branched at the tips, 2–3 branched at apex. *Pedicel* 20–26 × 4.5–6.5 µm (av. = 23 × 6 µm, n = 30), persistent, hyaline.

Materials examined: CHINA, Guizhou Province, Anshun City, Longdong Scenic Area (26°6′32.28″ N, 105°52′30.23″ E) III on *Koelreuteria bipinnata* Franch., July 10, 2021 Q.Z. Wu, GMB0105.

Notes: The leaves of *Koelreuteria bipinnata* have strong antimicrobial activity, and their extracts contain a variety of components that can inhibit bacteria and fungi [104,105]. We collected rust samples on the leaves of *Koelreuteria bipinnata*, and it can be identified as *Ny. koelreuferiae* based on both morphological and molecular evidences (Figure 5 and Figure 11). This rust fungus has already been reported in Zhejiang province, China in 1928 [106], and here we reported it on the same host species in Guizhou province. 

***Phragmidium cymosum*** Q. Z. Wu, T. Z. Liu, P, Zhao & Q. R. Li, **sp. nov**. Figure 12

MycoBank number: MB847102 

Etymology: Epithet follows to the epithet of host species, *Rosa cymosa* Tratt.

Holotype: GMB0108

*Parasitic* on the leaves of *Rosa cymosa*. *Uredinia Uredo*-type (*Uraecium* type II), hypophyllous, yellowish, aggregated, 0.1–0.4 mm (av. = 0.2 mm, n = 30), with thick-walled, gradual thickening by the roots of the paraphyses, incurved, intermixed paraphyses, 25–70 × 9–15 µm (av. = 44 × 11 µm, n = 30), wall 3–11 µm thick (av. = 7 µm, n = 30), top width 5–13 µm thick (av. = 8.5 µm, n = 30), *Urediniospores* borne singly, mostly echinulate, globose, broadly or obovoid, echinulate, 19–27 × 21–29 µm (av. = 24 × 26 µm, n = 30), cytoplasm orange, wall 1–2.5 µm thick (av. = 1.5 µm, n = 30). 

Materials examined: CHINA, Guizhou Province, Guiyang City, Campus of Guizhou Medical University (26°22′46.37″ N, 106°37′29.33″ E), II on *R. cymosa,* 6 October 2021, Q. Z. Wu, GY-XGQW GMB0108, holotype, CFSZ 50542, isotype; CHINA, Guizhou Province, Guiyang City, Campus of Guizhou Medical University (26°22′48.22″ N, 106°37′30.12″ E), II on *R. cymosa,* 6 October 2021, Q. Z. Wu, GMB0115.

Notes: *Phragmidium cymosum* formed a distinct lineage in the phylogenetic tree (Figure 3) with a high support rate (98/1). Morphologically, it differs from those species by its paraphyses with relatively thick wall, by the width of the urediniospores (21–29 µm in *P. cymosum*; 18–21 µm in *P. japonicum*; 11–18 µm in *P. jiangxiense*; 15–20 in *P. rosae-multiflorae*) [63,76]. In addition, the paraphyses of *P. cymosum* (25–70 × 9–15 µm) are thicker compared to those of *P. jiangxiense* (22–41 × 6–10 µm), and the cell walls of the paraphyses in *P. cymosum* (3–11 µm) are thicker than those of *P*. *rosae-multiflorae* (1 µm). *Phragmidium rosae-multiflorae* was once reported on the same host species in China [107].

The leaves, flowers and roots of *Rosa cymosa* can be used as the Chinese herbal medicine. Moreover, it has anti-inflammatory components, can be used to treat burns, analgesic [108,109,110]. 

***Phragmidium rosae-roxburghii*** J.E. Sun & Yong Wang bis Figure 13

MycoBank number: MB845041

*Parasitic* on the leave of *Rosa roxburghii* Tratt. *Uredinia Uredo*-type (*Uraecium* type II), hypophyllous, yellowish, aggregated, with thick-walled, gradual thickening by the roots of the paraphyses, incurved, intermixed paraphyses, 23–50 × 8–13.5 µm (av. = 36.5 × 10 µm, n = 30), wall 2–8.5 µm thick (av. = 5.5 µm, n = 30), *Urediniospores* borne singly, mostly echinulate, globose, broadly or obovoid, echinulate, 18–29 × 17–24 µm (av. = 24 × 20 µm, n = 30), cytoplasm orange, wall 0.5–1.5 µm thick (av. = 1 µm, n = 30).

Materials examined: CHINA, Guizhou province, Anshun City, Longdong Scenic Area (26°6′46.32″ N, 105°52′32.23″ E) II on *Rosa roxburghii* Tratt, 10 July 2021 Q.Z. Wu, GMB0104, CFSZ 50540.

Notes: *Rosa roxburghii* as a Chinese herbal medicine is used as a remedy for respiratory diseases. It has been recently reported to be an antioxidant and anticoagulant, and also be used to treat dyspepsia, dysentery, hypo immunity, and neurasthenia [65,111]. Our collection (GMB0104) and *P. rosae-roxburghii* form a clade in phylogenetic tree with a high support value (100/1). The morphological characteristics of GMB0104 are consistent with those of *P. rosae-roxburghii*. *Phragmidium rosae-roxburghii* was introduced based on the specimen collected from Guizhou province [65]. Here we reported it on medicinal plant in the same province.

***Phragmidium tormentillae*** Fuckel, Jb. nassau. Ver. Naturk. 23-24: 46 (1870) [1869-70] Figure 14

Mycobank number: MB177066

*Spermogonia* were not observed. *Uredinia* hypophyllous, scattered, subepidermal and erumpent becoming pulverulent, 0.5 mm diam. *Urediniospores* circular or obovoid, pale yellow, 18–24 × 14–20 µm (av. = 211 × 17 µm, n = 30), walls hyaline, echinulate, 1–1.5 mm thick.

Materials examined: CHINA, Guizhou Province, Qingzhen City, Xiasi Town (26°27′16.48″ N, 107°20′8.23″ E) II on *Potentilla simulatrix* Th. Wolf, 22 July 2021 Q.Z. Wu, GMB0114.

Notes: *Potentilla* is documented in most areas of China, which was used as a traditional Chinese medicine for hemostasis and treatment of malaria [112]. The pharmacological activities of *Potentilla* are mainly related to antioxidant, hypoglycemic, anti-inflammatory, antibacterial, antitumor, and cardiovascular system protective effects [113]. It also has positive effects on hemorrhagic cystitis [114]. Our collection on *Potentilla simulatrix* (GMB0110) is located in the same clade with *P. tormentillae* (Figure 3). Based on the size, shape and wall thickness of the urediniospores, as well as the presence of sparse spines on the surface of the urediniospores, new collection is identified as *P. tormentillae* [60,115,116]. However, no spermatozoa and paraphyses were observed in new collections. *Phragmidium tormentillae* was firstly collected in Norway in 1895 [116]. This is the first record for Chinese mainland.

## 4. Discussion

The research on rust diseases in medicinal plants has been relatively less compared to those in economic crops, mainly because in the past, medicinal plants were mostly sourced from the wild, with fewer incidents of rust diseases and thus had not received much attention. However, as the cultivation area of medicinal plants continues to increase, rust diseases have gradually become one of the important diseases affecting the quality and yield of medicinal materials. Attention on rust diseases in medicinal plants have been steadily increasing over time [17]. 

A total of 79 rust species were found to cause diseases on 76 species of medicinal plants from 33 families in China [16,17]. Rust diseases have become the primary diseases on some important medicinal plants in their primary growth regions, with strong prevalence and large damage areas, such as safflower rust disease (*Puccinia carthami*), Japanese yam rust (*Puccinia dioscoreae*), Radix glehniae rust (*Puccinia phellopteri*), and Rust on bulbus fritillariae ussuriensis (*Uromyces aecidiiformis*) [117,118,119,120]. Most researches on rusts on medicinal plants have been focused on the descriptions of symptoms, the incidence scopes and geographic distribution and the rough morphological descriptions of some spores [17]. However, high morphological variations, wide host range and complicated life cycles, identification of rust fungi is very difficult solely based on morphologies or host specificity. Herein, with the aid of morphological and molecular data, ten rust species have been found on medicinal plants collected from Guizhou province, including three new species and six known species. Among them, *Hamaspora rubi-alceifolii*, *Nyssopsora altissima* and *Phragmidium cymosum* were introduced as new to science. *Neophysopella ampelopsidis*, *Phragmidium tormentillae* was firstly introduced in Chinese mainland. *Melampsora laricis-populina*, *Melampsoridium carpini*, and *Nyssopsora koelreuteriae* were documented for the first time in Guizhou province. The accurate identification of rust fungi on medicinal plants will lay the foundation for disease control of medicinal plants.

The use of DNA sequences is becoming more and more important in the identification of rust fungi. Despite early research on rust fungi, the taxonomic system remains perplexing [121]. Distinguishing between individual rust fungi based solely on morphology is challenging [13,122,123]. Because the vast majority of rust fungi cannot be cultured on the artificial medium, pure culture strain cannot be obtained. Therefore, there is no enough DNA sequence available for a large number of rust species for a long time. As DNA extraction techniques continue to improve, valid DNA sequences will become increasingly available [61], and phylogenetic and morphological-based approaches will resolve the taxonomic confusion in rust fungi [11,124]. There were 337 species of rust fungi in 76 genera of 14 families using both morphological and molecular data from 86 natural reserves and national parks in the past five years [11]. Because molecular phylogenetic approaches can be used to connect the telial and aecial stages of rust fungi, they used more additional characters for species recognition [125]. Thus, their studies using DNA-based phylogenetic approach have facilitated precise identification of rust fungi at familial, generic, and species level. These studies can present a significant contribution to the knowledge of rust flora in China, especially those on medicinal plants.

## Figures and Tables

**Figure 1 jof-09-00953-f001:**
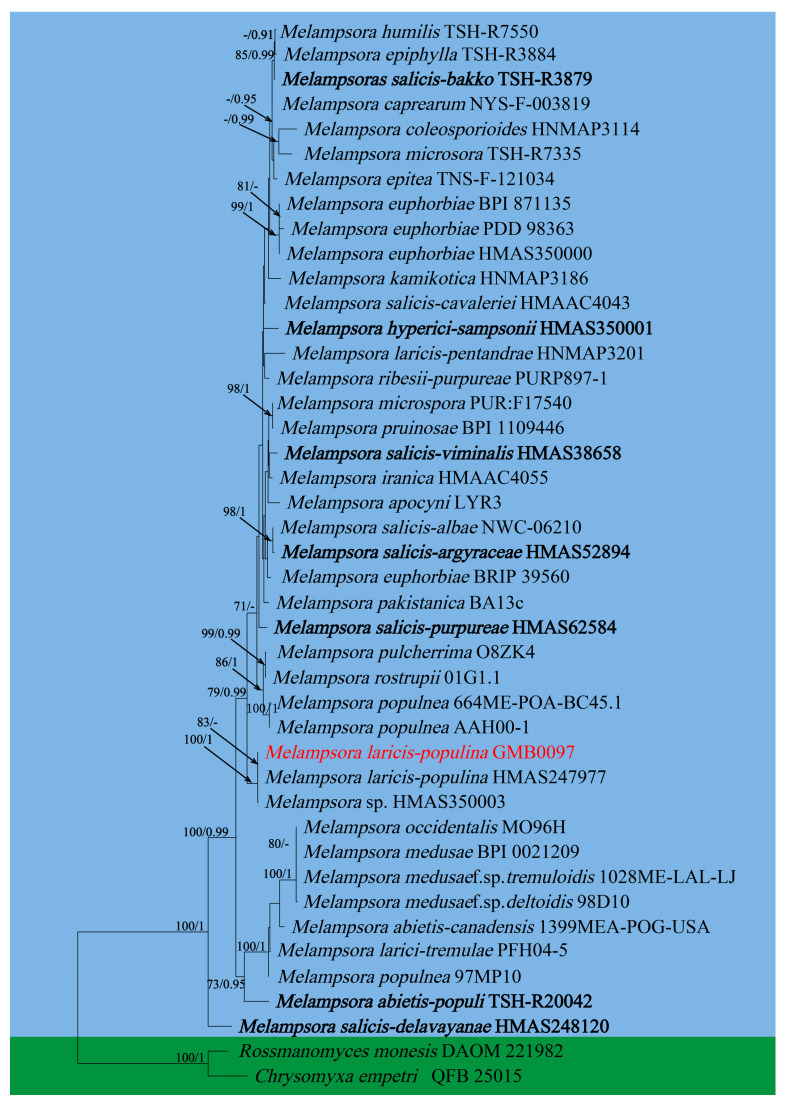
RAxML tree of the family *Melampsoraceae* based on rDNA ITS and LSU sequence. ML bootstrap supports (≥70%) and Bayesian posterior probability (≥0.90) are indicated as ML/BYPP. The tree is rooted to *Rossmanomyces monesis* and *Chrysomyxa empetri* [11]. The type specimens are shown as boldface. New sequences are in red.

**Figure 2 jof-09-00953-f002:**
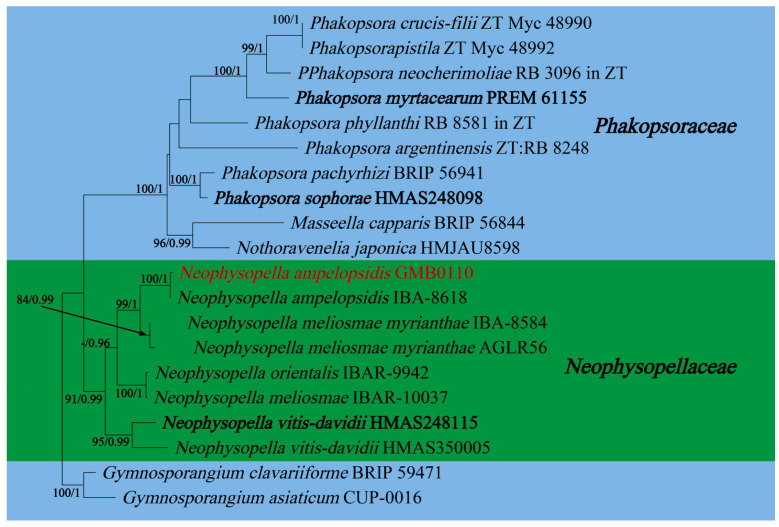
The RAxML tree of the family *Neophysopellaceae* and *Phakopsoraceae* based on rDNA ITS and LSU sequences. ML bootstrap supports (≥70%) and Bayesian posterior probability (≥0.90) are indicated as ML/BYPP. The tree is rooted to *G. asiaticum* and *G. clavariiforme* [11]. The type specimens are shown as boldface. New sequences are in red.

**Figure 3 jof-09-00953-f003:**
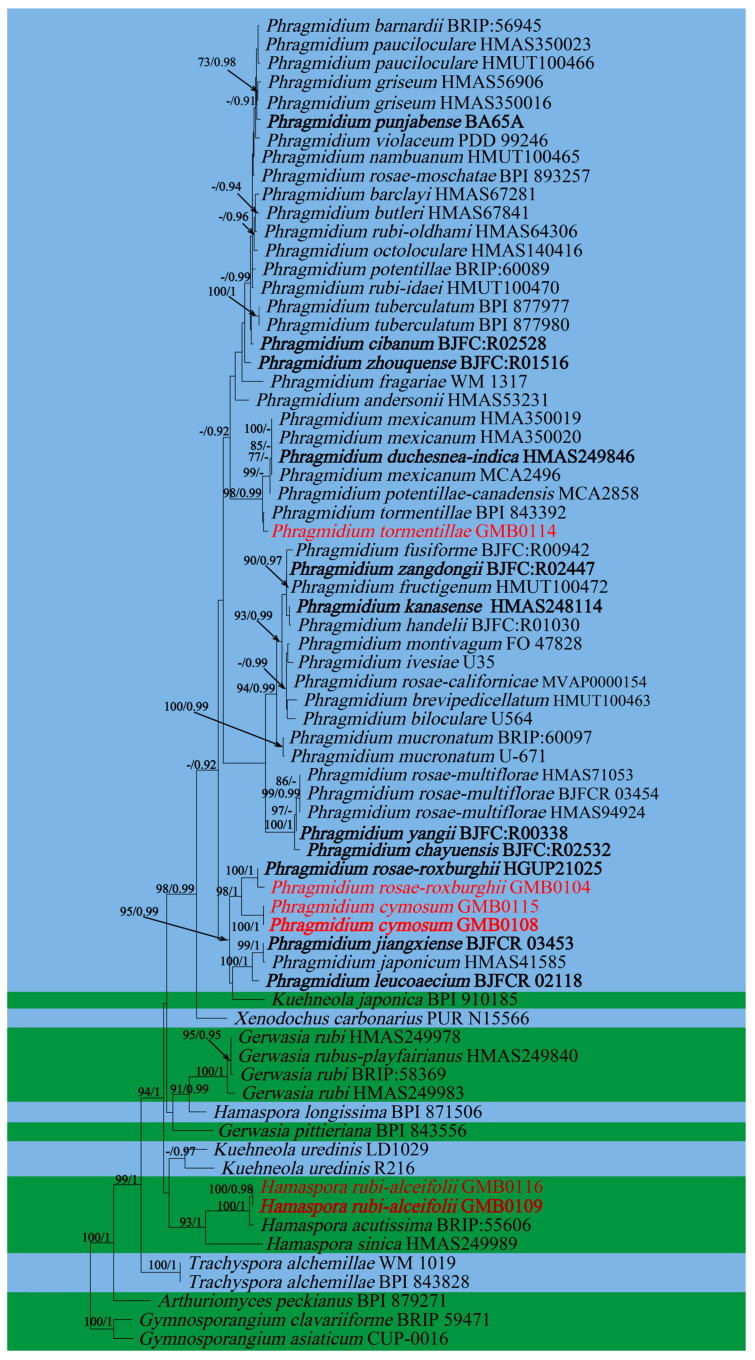
RAxML tree of the family *Phragmidiaceae* based on rDNA ITS and LSU sequences. ML bootstrap supports (≥70%) and Bayesian posterior probability (≥0.90) are indicated as ML/BYPP. The tree is rooted to *G. asiaticum* and *G. clavariiforme* [11]. The type specimens are shown as boldface. New sequences are in red.

**Figure 4 jof-09-00953-f004:**
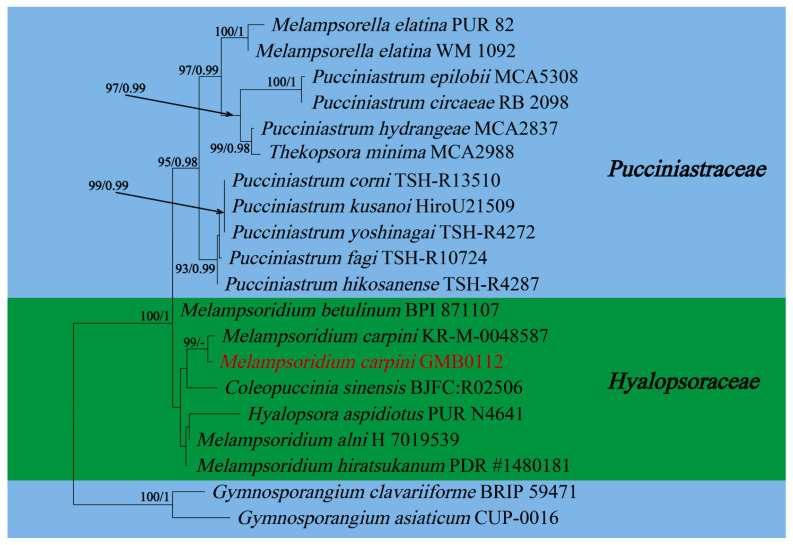
RAxML tree of the family *Pucciniastraceae* and *Hyalopsoraceae* based on rDNA ITS and LSU sequence. ML bootstrap supports (≥70%) and Bayesian posterior probability (≥0.90) are indicated as ML/BYPP. The tree is rooted to *G. asiaticum* and *G. clavariiforme* [11]. New sequences are in red.

**Figure 5 jof-09-00953-f005:**
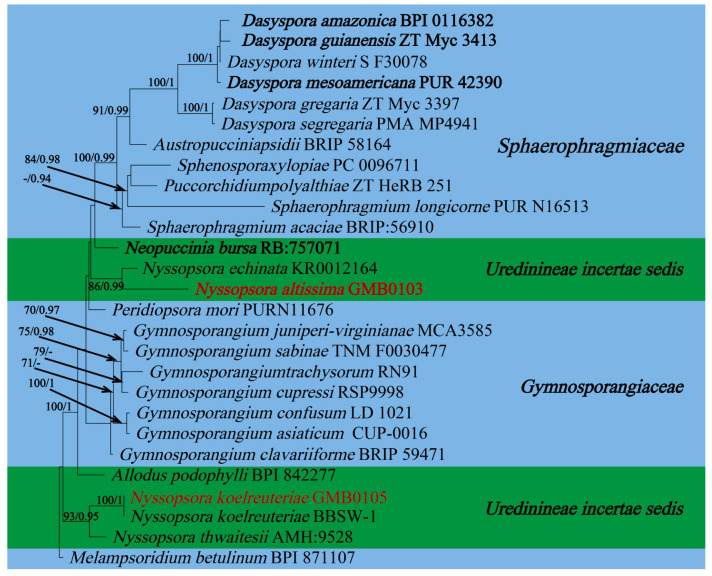
The RAxML tree of the families *Gymnosporangiaceae*, *Sphaerophragmiaceae* and *Uredinineae incertae sedis* based on rDNA ITS and LSU sequences. ML bootstrap supports (≥70%) and Bayesian posterior probability (≥0.90) are indicated as ML/BYPP. The tree is rooted to *Melampsoridium botulinum* [40]. The type specimens are shown as boldface. New sequences are in red.

**Figure 6 jof-09-00953-f006:**
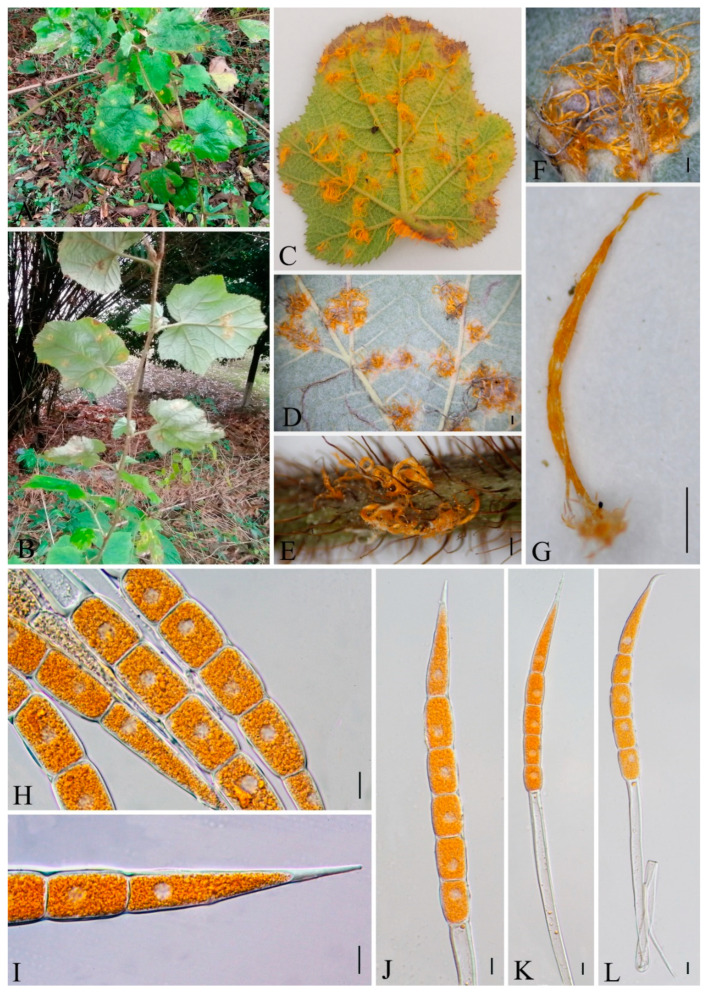
*Hamaspora rubi-alceifolii* (GMB0109). (**A**,**B**) host and its habitat; (**C**–**F**) telia on the hypophyllous leaf and stem; (**G**) teliospore cluster; (**H**–**L**) fusiform teliospores with 5–6 septa. Scale bars: (**D**) = 1 mm; (**E**–**G**) = 0.5 mm; (**H**–**L**) =10 µm.

**Figure 7 jof-09-00953-f007:**
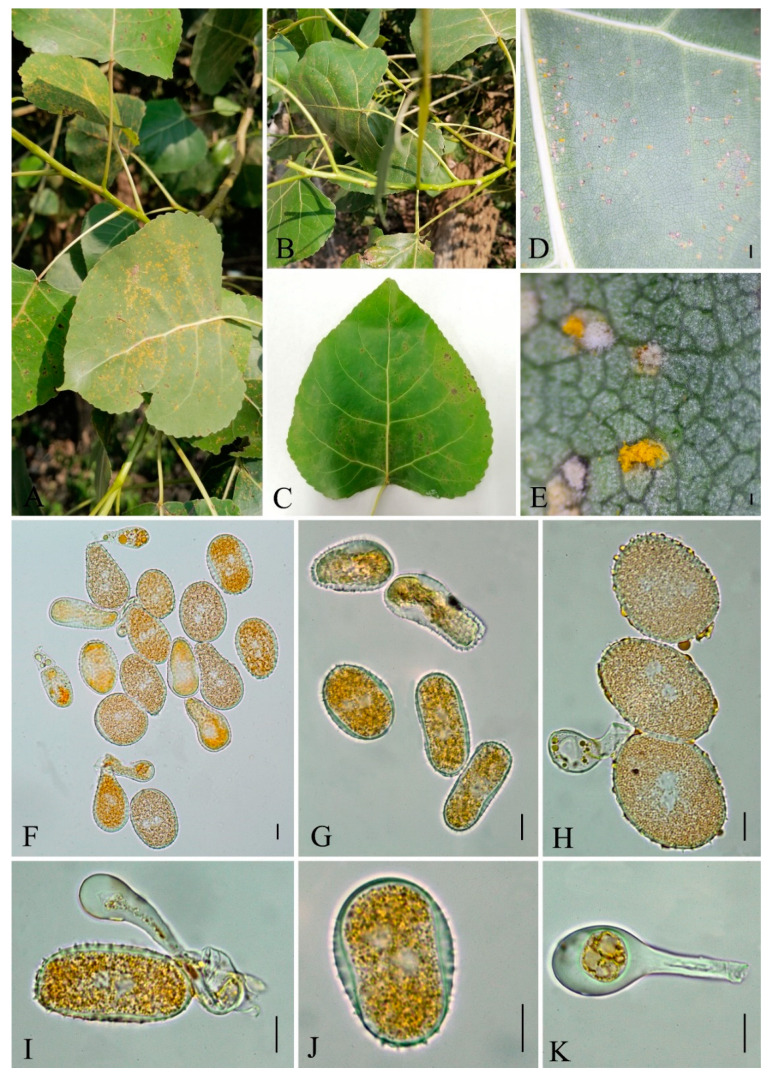
*Melampsora laricis-populina* (GMB0097) (**A**,**B**) host and its habitat; (**C**–**E**) uredinia; (**F**–**K**) urediniospores and paraphyses. Scale bars: (**D**) = 1 mm; (**E**) = 0.1 mm; (**F**–**K**) = 10 µm.

**Figure 8 jof-09-00953-f008:**
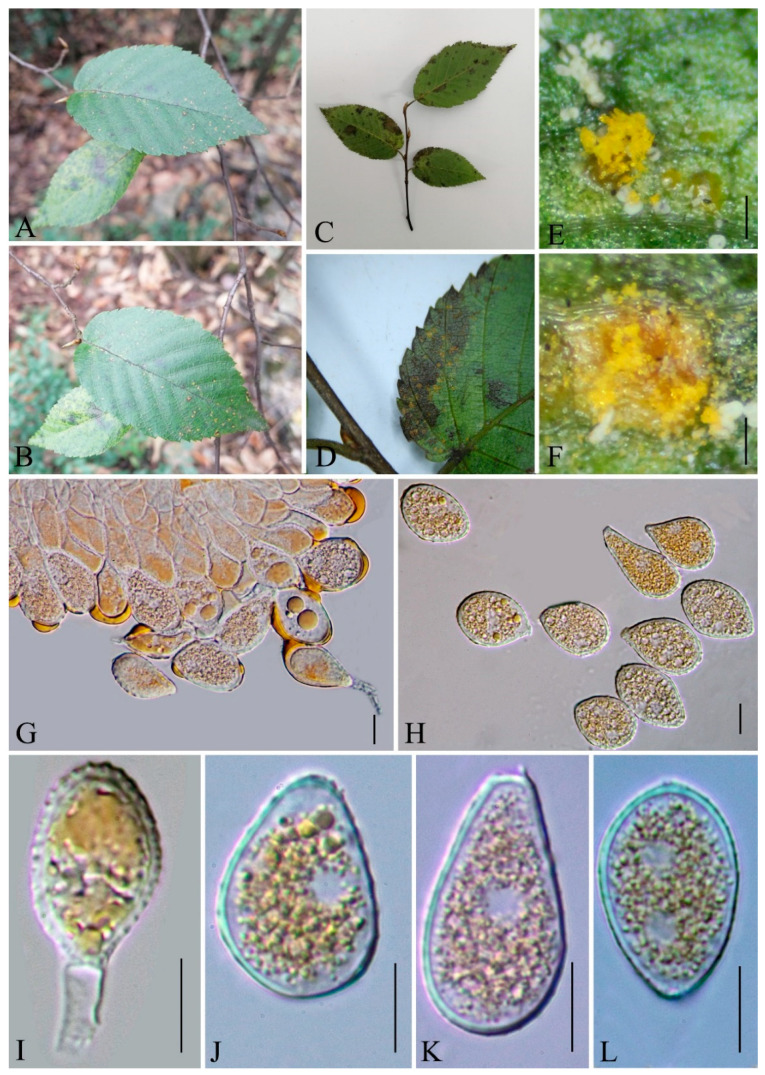
*Melampsoridium carpini* (GMB0112) (**A**,**B**) host; (**C**–**F**) uredinia; (**G**–**L**) urediniospores. Scale bars: (**E**,**F**) = 0.1 mm; (**G**–**L**) = 10 µm.

**Figure 9 jof-09-00953-f009:**
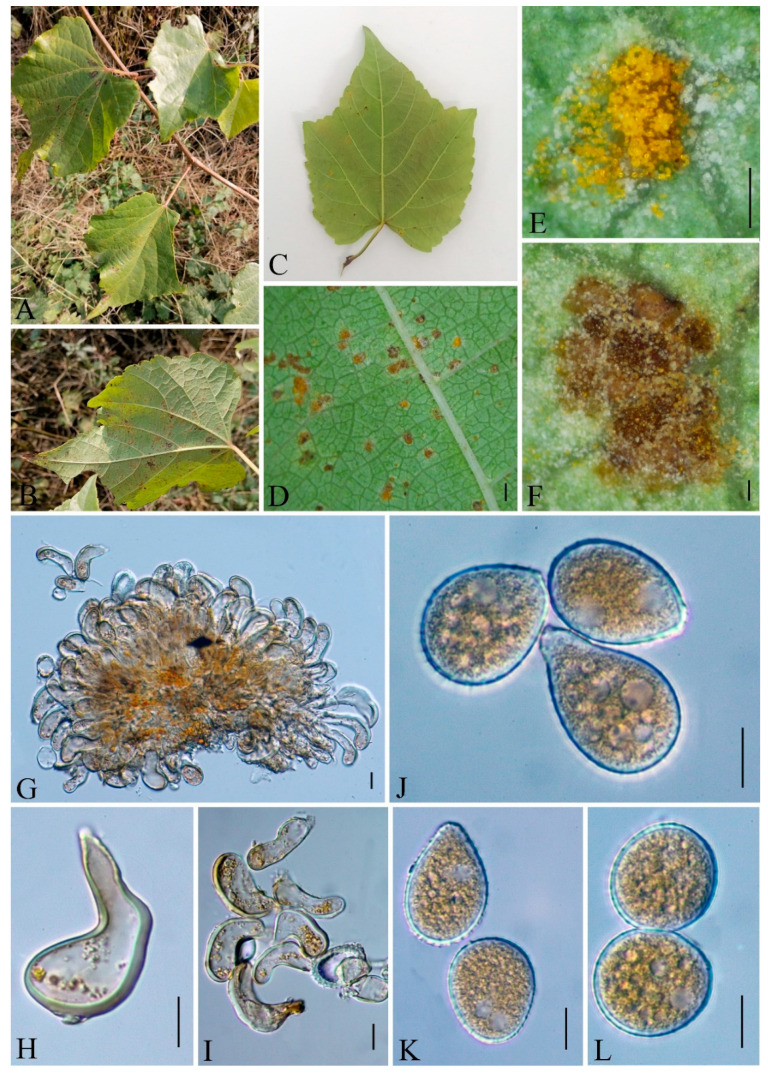
*Neophysopella ampelopsidis* (GMB0110). (**A**,**B**) host and its habitat; (**C**–**F**) uredinia.; (**G**–**L**) urediniospores and paraphyses. Scale bars: (**D**–**F**) = 0.1 mm, (**G**–**L**) = 10 µm.

**Figure 10 jof-09-00953-f010:**
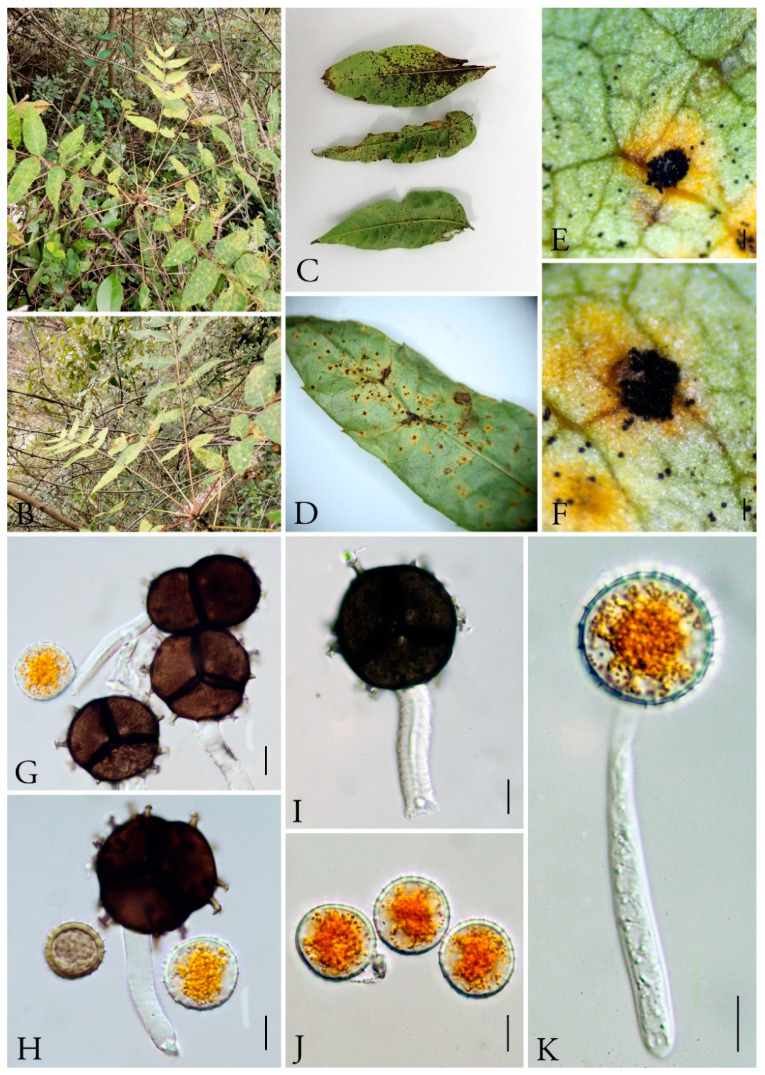
*Nyssopsora altissima* (GMB0103). (**A**,**B**) host and its habitat; (**C**,**D**) telia on the amphigenous leaf; (**E**,**F**) telia; (**G**–**I**) teliospores; (**J**,**K**) urediniospores with handle. Scale bars: (**E**,**F**) = 0.1 mm; (**G**–**K**) = 10 µm.

**Figure 11 jof-09-00953-f011:**
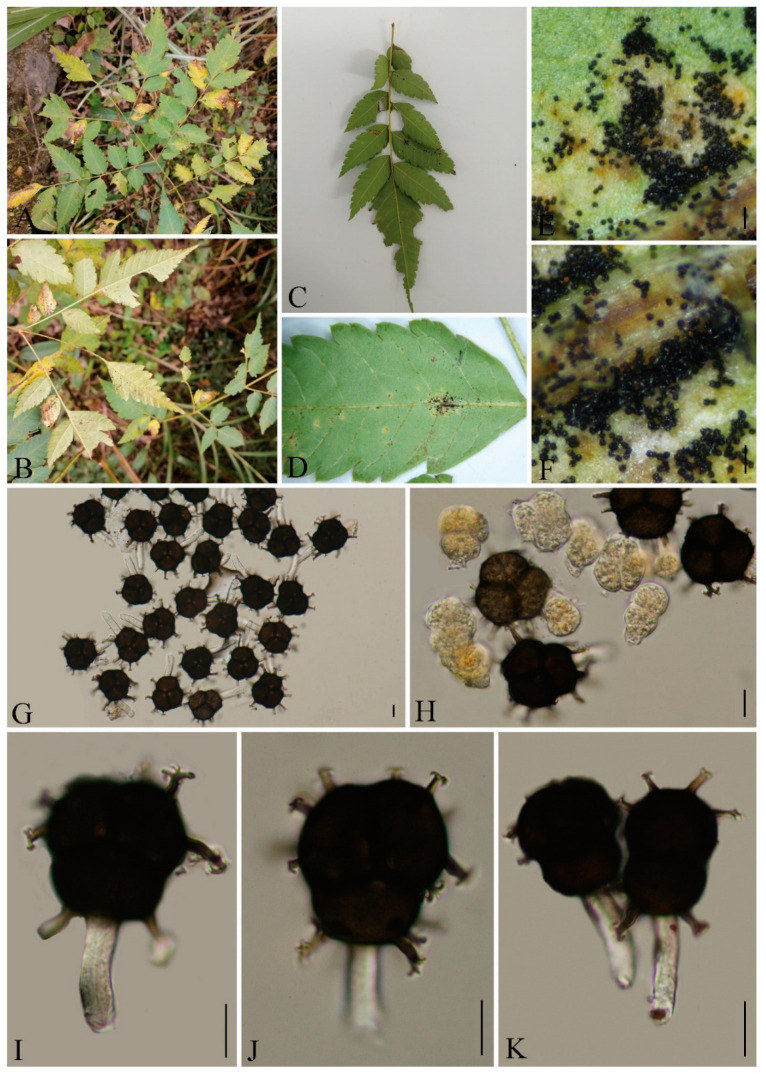
*Nyssopsora koelreuteriae* (GMB0105). (**A**,**B**) Host and its habitat. (**C**–**F**) Telia (**G**–**K**) Teliospores on the stem. Scale bars: (**E**,**F**) = 0.1 mm; (**G**–**K**) = 10 µm.

**Figure 12 jof-09-00953-f012:**
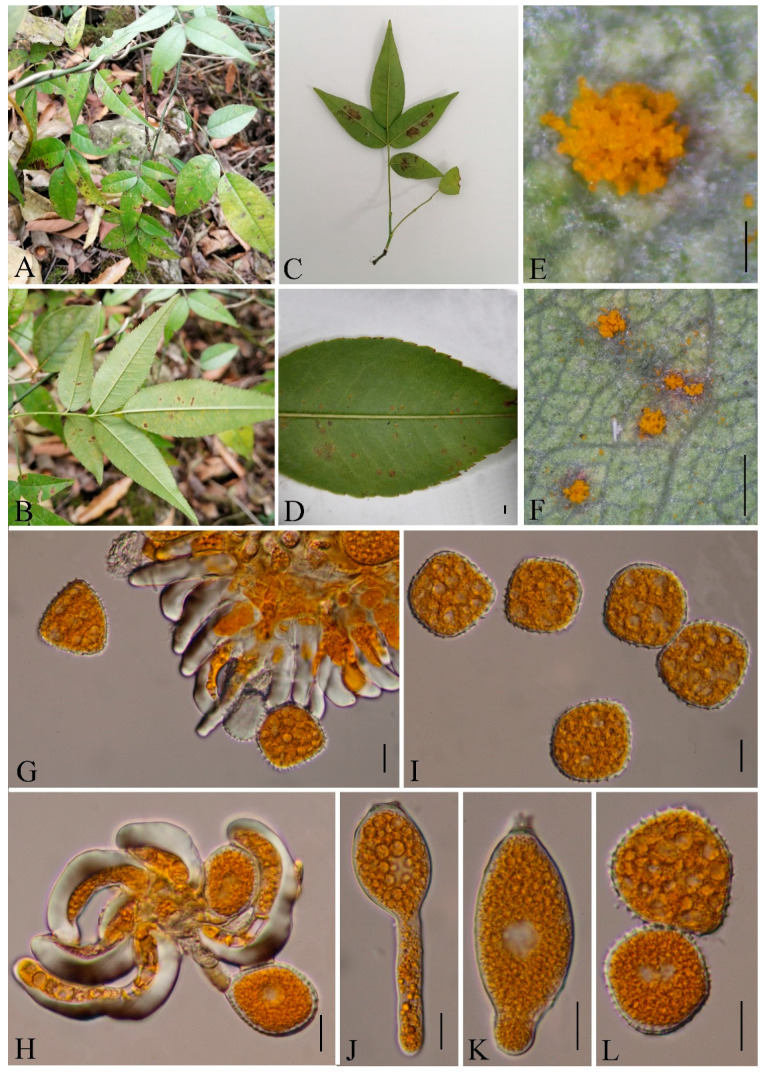
*Phragmidium cymosum* (GMB0108). (**A**,**B**) host and its habitat; (**C**,**D**) uredinia on the hypophyllous leaf surfaces; (**E**,**F**) uredinia; (**G**–**L**) globose or obovoid urediniospores with echinulate spines or with abnormal protrusions and paraphyses. Scale bars: (**D**) = 1 mm; (**E**) = 0.1 mm; (**F**) = 0.5 mm; (**G**–**L**) = 10 µm.

**Figure 13 jof-09-00953-f013:**
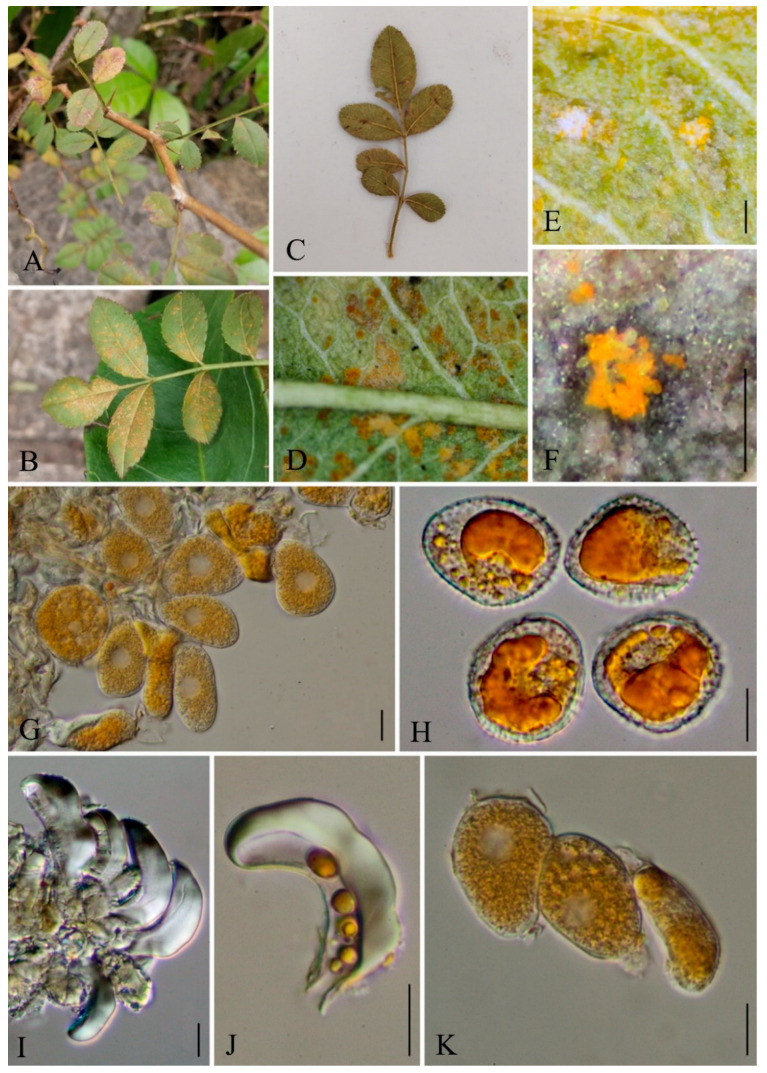
*Phragmidium rosae-roxburghii* (GMB0104). (**A**,**B**) host and its habitat; (**C**,**D**) uredinia on the hypophyllous leaf surfaces; (**E**,**F**) uredinia; (**G**–**K**) globose or oval urediniospores with echinulate spines and paraphyses. Scale bars: (**E**,**F**) = 0.5 mm, (**G**–**K**) = 10 µm.

**Figure 14 jof-09-00953-f014:**
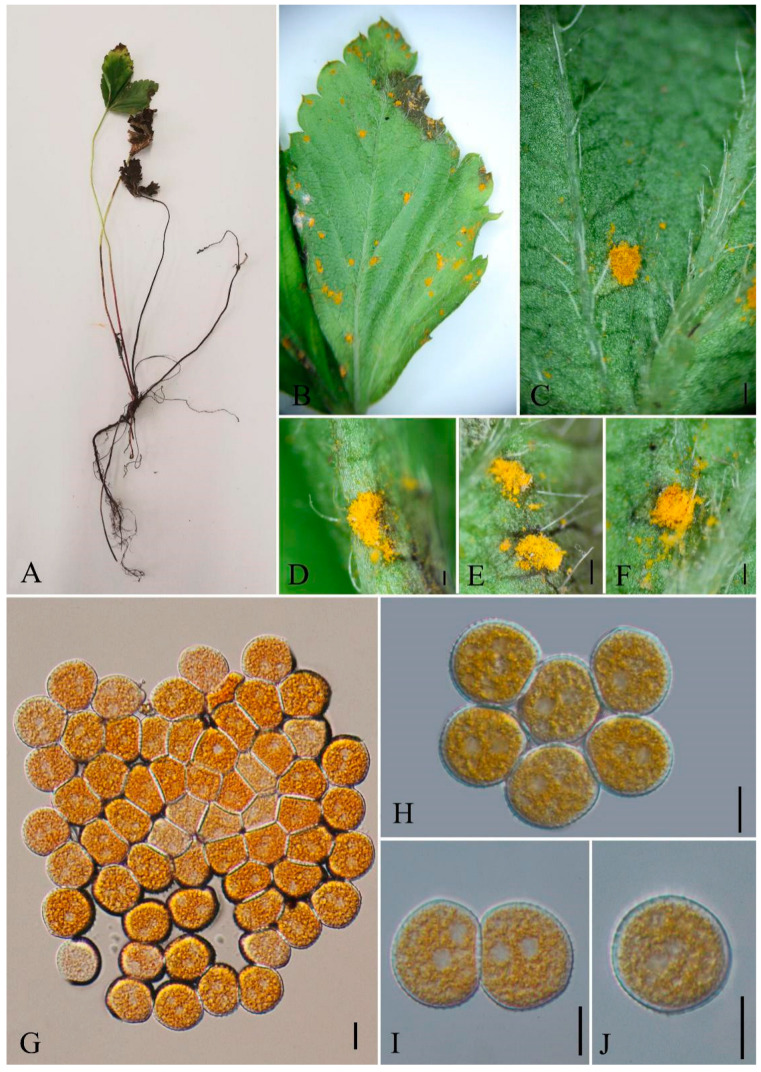
*Phragmidium tormentillae* (GMB0114). (**A**) host; (**B**–**F**) uredinia; (**G**–**J**) urediniospores. Scale bars: (**C**–**F**) = 0.1 mm; (**G**–**J**) = 10 µm.

**Table 1 jof-09-00953-t001:** GenBank accession number and information of taxa used for phylogenetic analyses.

Family Name	Species	Specimen No.	Host	Province, Country	GenBank Accession No.		Reference
					ITS	LSU	
*Chrysomyxaceae*	*Chrysomyxa empetri*	QFB 25015	*Empetrum nigrum*	Ste-Anne-des-Monts, Quebec, Canada	GU049434	GU049526	[36]
	*Rossmanomyces monesis*	DAOM 221982	*Moneses* (*=Pyrola*) *uniflora*	Graham Island, British Columbia, Canada	GU049476	GU049547	[36]
*Gymnosporangiaceae*	*Gymnosporangium asiaticum*	CUP-0016	*Juniperus chinensis*	Gifu prefecture, Japan	MN642593	MN642617	[37]
	*Gymnosporangium clavariiforme*	BRIP 59471	*Crataegus* sp.	—	—	MW049261	[10]
	*Gymnosporangium confusum*	LD 1021	*Crataegus monogyna*	Turkey	HM114219	HM114219	—
	*Gymnosporangium cupressi*	RSP9998	*—*	—	KJ720169	KJ720169	—
	*Gymnosporangium juniperi-virginianae*	MCA3585	*Cupressaceae juniperus*	Virginia, America	—	MG907217	[10]
	*Gymnosporangium sabinae*	TNM F0030477	*Pyrus communis*	Bulgaria: Sofia	KY964764	KY964764	[38]
	*Gymnosporangium trachysorum*	RN91	*—*	—	KJ720184	KJ720184	—
*Hyalopsoraceae*	*Coleopuccinia sinensis*	BJFC:R02506	*Cotoneaster microphyllus*	China	—	MF802285	[39]
	*Hyalopsora aspidiotus*	PUR N4641	*Gymnocarpium dryopteris*	China	—	MW049264	[40]
	*Melampsoridium alni*	H 7019539	*Alnus mandshurica*	Finland	KF031557	KF031534	[41]
	*Melampsoridium betulinum*	BPI 871107 (MCA2884)	*Alnus* sp.	Costa Rica	—	DQ354561	[8]
	*Melampsoridium hiratsukanum*	PDR #1480181	*Alnus rhombifolia*	USA: Santa Cruz County, California	KC313888	KC313888	[42]
	*Melampsoridium carpini*	KR-M-0048587	*Carpinus betulus*	—	MH908486	MH908486	[43]
	*Melampsoridium carpini*	GMB0112	*Carpinus turczaninowii*	Guizhou, China	OQ067091	—	This study
*Melampsoraceae*	*Melampsora abietis-canadensis*	1399MEA-POG-USA	*Populus grandidentata*	Wisconsin, America	JN881733	JN934918	[44]
	*Melampsora abietis-populi*	TSH-R20042 *	*Populus yunnanensis*	Yunnan, China	JN881739	JN934933	[44]
	*Melampsora apocyni*	LYR3	*Apocynum venetum*	Xinjiang, China	KR296802	KR296803	—
	*Melampsora caprearum*	NYS-F-003819	*Salix caprea*	Germany	KU550034	KU550033	[45]
	*Melampsora coleosporioides*	HNMAP3114	*Salix* sp.	Japan	KF780755	KF780638	[46]
	*Melampsora epiphylla*	TSH-R3884	*Salix bakko*	Japan	KF780787	KF780670	[45]
	*Melampsora epitea*	TNS-F-121034	*Salix viminalis*	Germany	KX386070	KX386097	[35]
	*Melampsora euphorbiae*	HMAS350000	*Euphorbia kansui*	Guangxi, China	MK518875	MK518545	[11]
	*Melampsora euphorbiae*	BRIP 39560	*Euphorbia peplus*	Queensland, Australia	—	EF192199	[40]
	*Melampsora euphorbiae*	BPI 871135 (U681)	*Euphorbia heterophylla*	Oman	—	DQ351722	[47]
	*Melampsora humilis*	TSH-R7550	*Salix koriyanagi*	Miyagi, Japan	KF780812	—	[46]
	*Melampsora hyperici-sampsonii*	HMAS350001 *	*Hypericum sampsonii*	Guangxi, China	MK518877	MK518547	[11]
	*Melampsora iranica*	HMAAC4055	*Salix* sp.	China	MK372158	MK372191	[48]
	*Melampsora kamikotica*	HNMAP3186	*Chosenia arbutifolia*	China	KF780760	KF780643	[46]
	*Melampsora laricis-pentandrae*	HNMAP3201	*Salix pentandra*	Inner Mongolia, China	KF780801	KF780684	[46]
	*Melampsora laricis-populina*	HMAS247977	*Populus simonii*	Haixi Qinghai, China	MK028583	MK064524	[49]
	*Melampsora laricis-populina*	GMB0097	*Populus lasiocarpa*	Guizhou, China	OQ067085	—	This study
	*Melampsora larici-tremulae*	PFH04-5	*Populus tremula*	France	JN881744	JN934956	[44]
	*Melampsora medusae*	BPI 0021209	*—*	America	JX416848	JX416843	[44]
	*Melampsora medusae* f.sp. *deltoidis*	98D10	*Populus* × *euramericana*	South Africa	GQ479307	JN934962	[44]
	*Melampsora medusae* f.sp. *tremuloidis*	1028ME-LAL-LJ	*Larix laricina*	Quebec, Lac Saint Jean, Peribonka, Canada	GQ479883	JN934965	[44]
	*Melampsora microsora*	TSH-R7335	*Salix subfragilis*	Hiroshima, Japan	KF780833	KF780730	[46]
	*Melampsora microspora*	PUR:F17540	*populus nigra*	Iraq	JN881737	JN934931	[44]
	*Melampsora occidentalis*	MO96H	*Populus trichocarpa*	Idaho, America	JN881740	JN934934	[44]
	*Melampsora pakistanica*	BA13c	*Euphorbia helioscopia*	Pakistan	KX237555	KX237556	[50]
	*Melampsora populnea*	664ME-POA-BC45.1	*Populus alba*	British Columbia, Canada	EU808021	FJ666510	[51]
	*Melampsora populnea*	97MP10	*—*	France	EU808035	FJ666523	[51]
	*Melampsora populnea*	AAH00-1	*Populus alba*	England	AY444772	AY444786	[52]
	*Melampsora pruinosae*	BPI 1109446	*Populus diversifolia*	Xinjiang Uygur Zizhiqu, Shule, China	GQ479899	JN934938	[44]
	*Melampsora pulcherrima*	O8ZK4	*Mercurialis annua*	Italy	GQ479320	JN934941	[44]
	*Melampsora ribesii-purpureae*	PURP897-1	*Salix purpurea*	—	AY444770	AY444791	[52]
	*Melampsora euphorbiae*	PDD 98363	*Ricinus communis*	—	—	KJ716352	[36]
	*Melampsora rostrupii*	01G1.1	*Mercurialis perennis*	France	EU808038	JN934942	[44]
	*Melampsora salicis-albae*	NWC-06210	*Salix alba*	Rothamsted, England	KF780757	KF780640	[46]
	*Melampsora salicis-argyraceae*	HMAS52894 *	*Salix argyracea*	Xinjiang, China	KF780733	KF780616	[46]
	*Melampsora salicis-bakko*	TSH-R3879 *	*Salix bakko*	—	KC631854	KC685611	[45]
	*Melampsora salicis-cavaleriei*	HMAAC4043	*Salix serrulatifolia*	China	MK277296	MK277301	[48]
	*Melampsora salicis-delavayanae*	HMAS248120 *	*Salix delavayana*	Yunnan, China	MK518954	MK518651	[11]
	*Melampsora salicis-purpureae*	HMAS62584 *	*Salix purpurea*	Shandong, China	KF780766	KF780649	[46]
	*Melampsora salicis-viminalis*	HMAS38658 *	*Salix viminalis*	Tibet, China	KF780732	KF780615	[46]
	*Melampsora* sp.	HMAS350003		Xinjiang, China	MK518844	MK518499	[11]
*Neophysopellaceae*	*Neophysopella ampelopsidis*	IBA-8618	*Ampelopsis brevipedunculata*	Kagoshima, Japan	AB354774	AB354741	[11]
	*Neophysopella ampelopsidis*	GMB0110	*Ampelopsis sinica*	Guizhou, China	OQ067090	—	This study
	*Neophysopella meliosmae*	IBAR-10037	*Meliosma myriantha*	Ibaraki, Japan	KC815591	KC815650	[53]
	*Neophysopella meliosmae-myrianthae*	AGLR56	—	—	MK290819	MK290819	—
	*Neophysopella meliosmae-myrianthae*	IBA-8584	*Vitis coignetiae*	Tochigi, Japan	AB354785	AB354748	[53]
	*Neophysopella orientalis*	IBAR-9942	*Meliosma tenuis*	Tochigi, Japan	KC815597	KC815656	[54]
	*Neophysopella vitis-davidii*	HMAS350005	*Viola faurieana*	Chongqing, China	MK518870	MK518536	[11]
	*Neophysopella vitis-davidii*	HMAS248115 *	*Vitis davidii*	Yunnan, China	—	MK518593	[11]
*Phakopsoraceae*	*Masseeëlla capparis*	RIP 56844	*Flueggea virosa*	Northern Territory, Australia	JX136798	JX136798	[13]
	*Nothoravenelia japonica*	HMJAU8598	—	China	—	MK296509	—
	*Phakopsora argentinensis*	ZT:RB 8248	*Croton* cf. *anisodontus*	—	KF528009	KF528009	[55]
	*Phakopsora crcis-filii*	ZT Myc 48990	*Annona paludosa*	Sinnamary, French Guiana	KF528016	KF528016	[55]
	*Phakopsora myrtacearum*	PREM 61155 *	*Eucalyptus grandis*	Maragua, Kenya	NR_132913	KP729473	[56]
	*Phakopsora neocherimoliae*	RB 3096 in ZT	*Annona cherimola*	San José Costa Rica	KF528011	KF528011	[55]
	*Phakopsora pachyrhizi*	BRIP 56941	*Neonotonia wightii*	Warrumbungle, New South Wales, Australia	—	KP729475	[56]
	*Phakopsora phyllanthi*	RB 8581	*Phyllanthus acidus*	Ceará, Brazil	KF528025	KF528025	[55]
	*Phakopsora pistila*	ZT Myc 48992	*Annona sericea*	French GuianaIracubo, French Guiana	KF528026	KF528026	[55]
	*Phakopsora sophorae*	HMAS248098 *	*Leptopus chinensis*	Beijing, China	—	MK518628	[11]
*Phragmidiaceae*	*Arthuriomyces peckianus*	BPI 879271	*Rubus occidentalis*	New York, America	GU058010	GU058010	[57]
	*Gerwasia pittieriana*	BPI 843556	*Rubus* sp.	—	KY764065	KY764065	—
	*Gerwasia rubi*	HMAS249978	*Rubus parkeri*	Yunnan, China	MK519039	MK518737	[11]
	*Gerwasia rubi*	HMAS249983	*Rubus setchuenensis*	Fujian, China	—	MK518734	[11]
	*Gerwasia rubi*	BRIP:58369	*Rubus* sp.	—	—	KT199397	[13]
	*Gerwasia rubi-playfairiani*	HMAS249840 *	*Rubus playfairianus*	Guangxi, China	MK518976	—	[11]
	*Hamaspora acutissima*	BRIP:55606	*Rubus rolfei*	Philippines	—	KT199398	[13]
	*Hamaspora rubi-alceifolii*	GMB0116	*Rubus alceaefolius*	Guizhou, China	—	OQ067533	This study
	*Hamaspora rubi-alceifolii*	GMB0109 *	*Rubus alceaefolius*	Guizhou, China	—	OQ067532	This study
	*Hamaspora longissima*	BPI 871506	*Rubus rigidus*	Eastern Cape, South Africa	—	MW049262	[40]
	*Hamaspora sinica*	HMAS249989	*Rubus setchuenensis*	Guangdong, China	MK519049	MK518636	[11]
	*Kuehneola japonica*	BPI 910185	*Rosa* sp.	—	KY764067	—	—
	*Kuehneola uredinis*	LD1029	*Rubus* sp.	New York, America	GU058013	GU058013	[57]
	*Kuehneola uredinis*	R216	*Rubus fruticosus*	Belgium	EU014068	EU014068	[58]
	*Phragmidium andersonii*	HMAS53231	*Potentilla fruticosa*	China	—	MG669120	[59]
	*Phragmidium barclayi*	HMAS67281	*Rubus austrotibetanus*	China	—	MG669117	[59]
	*Phragmidium barnardii*	BRIP:56945	*Rubus multibracteatus*	Queensland, Australia:	—	KT199402	[13]
	*Phragmidium biloculare*	U564	*Potentilla flabellifolia*	Washington, America	—	JF907670	[60]
	*Phragmidium brevipedicellatum*	HMUT100463	*Potentilla multifida*	Xinjiang, China	—	KU059170	—
	*Phragmidium butleri*	HMAS67841	*Rosa macrophylla*	China	—	MG669118	[59]
	*Phragmidium chayuensis*	BJFC:R02532 *	*Rosa duplicata*	China	MH128374	NG_064492	[59]
	*Phragmidium cibanum*	BJFC:R02528 *	*Rubus niveus*	China	MH128370	NG_064491	[59]
	*Phragmidium cymosum*	GMB0115	*Rosa cymosa*	Guizhou, China	OQ067097	OQ067531	This study
	*Phragmidium cymosum*	GMB0108 *	*Rosa cymosa*	Guizhou, China	OQ067096	OQ067530	This study
	*Phragmidium duchesnea-indica*	HMAS249846 *	*Duchesnea indica*	Yunnan, China	—	MK518681	[11]
	*Phragmidium fragariae*	WM 1317	*Potentilla sterilis*	—	—	AF426217	[61]
	*Phragmidium fructigenum*	HMUT100472	*Rosa glomerata*	Chongqing, China	—	KU059168	—
	*Phragmidium fusiforme*	BJFC:R00942	*Rosa hugonis*	China	—	KP407632	[62]
	*Phragmidium griseum*	HMAS56906	*Rubus crataegifolius*	Beijing, China	MH128377	MG669115	[59]
	*Phragmidium griseum*	HMAS350016	*Rosa* sp.	Beijing, China	—	MK518530	[11]
	*Phragmidium handelii*	BJFC:R01030	*Rosa webbiana*	China	—	KP407631	[62]
	*Phragmidium ivesiae*	U35	*Potentilla gracilis*	Utah, America	—	JF907672	[60]
	*Phragmidium japonicum*	HMAS41585	*Rosa laevigata*	Fujian, China	MN264716	MN264734	[63]
	*Phragmidium jiangxiense*	BJFCR 03453 *	*Rosa laevigata*	Jiangxi, China	MN264715	MN264733	[63]
	*Phragmidium kanasense*	HMAS248114 *	*Rosa fedtschenkoana*	Xinjiang, China	—	MK518464	[11]
	*Phragmidium leucoaecium*	BJFCR 02118 *	*Rosa* sp.	Yunnan, China	MN264719	MN264737	[63]
	*Phragmidium mexicanum*	HMAS350019	*Phoenix acaulis*	Yunnan, China	MK518980	MK518678	[11]
	*Phragmidium mexicanum*	HMAS350020	*Duchesnea indica*	Yunnan, China	MK518982	MK518680	[11]
	*Phragmidium mexicanum*	MCA2496	*Potentilla indica*	Maryland, America	—	JF907660	[60]
	*Phragmidium montivagum*	FO 47828	*Rosa* cf. *woodsii*	—	—	AF426213	[61]
	*Phragmidium mucronatum*	U-671	*Rosa* sp.	Oman	—	HQ412646	[64]
	*Phragmidium mucronatum*	BRIP:60097	*Rosa rubiginosa*	—	—	MW049275	[40]
	*Phragmidium nambuanum*	HMUT100465	*Rubus innominatus*	Chongqing, China	—	KU059165	—
	*Phragmidium octoloculare*	HMAS140416	*Rubus biflorus*	China	MH128376	MG669119	[59]
	*Phragmidium pauciloculare*	HMUT100466	*Rubus innominatus*	Chongqing, China	—	KU059162	—
	*Phragmidium pauciloculare*	HMAS350023	*Rubus corchorifolius*	Guangxi, China	MK518874	MK518542	[11]
	*Phragmidium potentillae*	BRIP:60089	*Acaena novae-zelandiae*	Tasmania, Australia	—	KT199403	[13]
	*Phragmidium potentillae-canadensis*	MCA2858	*Potentilla* sp.	New York, America	—	JF907666	[60]
	*Phragmidium punjabense*	BA65A *	*Rosa* sp.	Murree, Ghora Gali, Pakistan	KX358856	KX358854	[50]
	*Phragmidium rosae-californicae*	MVAP0000154	*Rosa californica*	Putah Creek Reserve, Davis, California, America	MK045315	MK045315	—
	*Phragmidium rosae-moschatae*	BPI 893257	*Rosa macrophylla*	—	—	KY798368	—
	*Phragmidium rosae-multiflorae*	HMAS71053	*Rosa multiflora*	Shanxi, China	—	KU059174	[65]
	*Phragmidium rosae-multiflorae*	HMAS94924	*Rosa multiflora*	Zhejiang, China	—	KU059175	[65]
	*Phragmidium rosae-multiflorae*	BJFCR 03454 *	*Rosa multiflora*	Jiangxi, China	MN264721	MN264739	[63]
	*Phragmidium rosae-roxburghii*	HGUP21025 *	*Rosa roxburghii*	Guizhou, China	OL684818	OL684831	[65]
	*Phragmidium rosae-roxburghii*	GMB0104	*Rosa xanthina*	Guizhou, China	OQ067092	—	This study
	*Phragmidium rubi-idaei*	HMUT100470	*Rubus saxatilis*	Chongqing, China	—	KU059163	—
	*Phragmidium rubi-oldhami*	HMAS64306	—	China	—	MG669116	[59]
	*Phragmidium tormentillae*	GMB00114	*Potentilla simulatrix*	Guizhou, China	OQ067093	—	This study
	*Phragmidium tormentillae*	BPI 843392	*Duchesnea* sp.	Maryland, America	—	DQ354553	[8]
	*Phragmidium tuberculatum*	BPI 877977	*Rosa floribunda*	Massachusetts, America	—	KJ841923	[66]
	*Phragmidium tuberculatum*	BPI 877980	*Rosa floribunda*	Oregon, America	—	KJ841922	[66]
	*Phragmidium violaceum*	PDD 99246	*Rubus* sp.	New Zealand	—	KJ716351	[36]
	*Phragmidium yangii*	BJFC:R00338 *	*Rosa* sp.	China	—	NG_060138	[62]
	*Phragmidium zangdongii*	BJFC:R02447 *	*Rosa tibetica*	Tibet, China	MH128372	NG_064490	[59]
	*Phragmidium zhouquense*	BJFC:R01516 *	*Rosa omeiensis*	China	—	NG_060139	[62]
	*Trachyspora alchemillae*	WM 1019	*Alchemilla vulgaris*	—	—	AF426220	[61]
	*Trachyspora alchemillae*	BPI 843828	*Alchemilla vulgaris*	Switzerland	—	DQ354550	[8]
	*Xenodochus carbonarius*	PUR N15566	*Sanguisorba officinalis*	—	—	MW049289	[40]
*Pucciniastraceae*	*Melampsorella elatina*	WM 1092	*Abies alba*	—	—	AF426232	[61]
	*Melampsorella elatina*	PUR 82	*Cerastium fontanum*	Minnesota, America	—	MG907233	[10]
	*Pucciniastrum circaeae*	RB 2098	*Circaea lutetiana*	—	—	AF426227	[61]
	*Pucciniastrum corni*	TSH-R13510	*Cornus kuosa*	Tottori, Japan	AB221437	AB221409	[67]
	*Pucciniastrum epilobii*	PUR N11088(MCA5308)	*Epilobium angustifolium*	—	—	MW147083	[40]
	*Pucciniastrum fagi*	TSH-R10724	*Fagus crenata*	Tochigi, Japan	AB221425	AB221378	[67]
	*Pucciniastrum hikosanense*	TSH-R4287 (IBA2565)	*Acer rufinerve*	Yamanashi Japan	AB221441	AB221388	[67]
	*Pucciniastrum hydrangeae*	MCA2837	*Hydrangea arborescens*	North Carolina, America	—	MG907240	[10]
	*Pucciniastrum kusanoi*	HiroU21509	*Clethra barbinervis*	Miyazaki, Japan	AB221426	AB221402	[67]
	*Pucciniastrum yoshinagai*	TSH-R4272 IBA8430	*Stewartia monadelpha*	Nara, Yoshino-gun, Japan	AB221434	AB221411	[67]
	*Thekopsora minima*	MCA2988	*Vaccinium angustifolium*	Maryland, America	—	MG907243	[10]
*Sphaerophragmiaceae*	*Austropuccinia psidii*	BRIP:58164	*Rhodamnia angustifolia*	Brisbane, Queensland. Australia	—	KF318449	[67]
	*Dasyspora amazonica*	BPI US0116382 *	*Xylopia amazonica*	Brazil	JF263460	JF263460	[68]
	*Dasyspora gregaria*	ZT Myc 3397	*Xylopia cayennensis*	French Guiana	JF263477	JF263477	[68]
	*Dasyspora guianensis*	ZT Myc 3413 *	*Xylopia benthamii*	French Guiana	JF263479	JF263479	[68]
	*Dasyspora mesoamericana*	PUR 42390 *	*Xylopia frutescens*	Panama	JF263480	JF263480	[68]
	*Dasyspora segregaria*	PMA MP4941	*Xylopia aromatica*	Panama	JF263488	JF263488	[68]
	*Dasyspora winteri*	S F30078	*Xylopia sericea*	Brazil	JF263492	JF263492	[68]
	*Puccorchidium polyalthiae*	ZT HeRB 251	*Polyalthia longifolia*	India	JF263493	JF263493	[68]
	*Sphaerophragmium acaciae*	BRIP:56910	*Albizzia* sp.	Kununurra, Western Australia, Australia	—	KJ862350	[69]
	*Sphaerophragmium longicorne*	PUR N16513	*Dalbergia hostilis*	—	—	MW147053	[40]
	*Sphenorchidium xylopiae*	PC 0096711	*Xylopia aethiopica*	western Central Africa.	KM217355	KM217355	[70]
*Uredinineae incertae sedis*	*Allodus podophylli*	BPI 842277	*Podophyllum peltatum*	Maryland, America	DQ354543	DQ354543	[67]
	*Neopuccinia bursa*	RB 75707	*Protium heptaphyllum*	Brazil	—	MH047186	—
	*Nyssopsora altissima*	GMB0103 *	*Ailanthus altissima*	Guizhou, China	OQ067089	OQ067529	This study
	*Nyssopsora echinata*	KR0012164	*Meum athamanticum*	—	—	MW049272	[40]
	*Nyssopsora koelreuteriae*	BBSW-1	*Koelreuteria bipinnata*	—	KT750965	KT750965	[71]
	*Nyssopsora koelreuteriae*	GMB0105	*Koelreuteria bipinnata*	Guizhou, China	OQ067088	—	This study
	*Nyssopsora thwaitesii*	AMH:9528	*Schefflera wallichiana*	India	KF550283	—	[72]

Notes: —: no data available; *: type specimens.

## Data Availability

All newly generated sequences were deposited in GenBank (https://www.ncbi.nlm.nih.gov/genbank/, accessed on 16 December 2022; Table 1). All new taxa were deposited in MycoBank (https://www.mycobank.org/, accessed on 3 January 2023; MycoBank identifiers follow new taxa).

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
