# Peer review of "Rust Fungi on Medicinal Plants in Guizhou Province with Descriptions of Three New Species"

_jof, 2023, doi:10.3390/jof9090953_

Round 1

Reviewer 1 Report

This work contributes many data to well understanding of the diversity of rust fungi on medicinal plant in China After review this manuscript and related documents, some problems and mistakes were found. The comments and revision parts of this manuscript are listed as following:

1. Some mistakes in format, text and references have been found, especially the reference in the attachment. I listed some of them in the attached file and marked in yellow color. Please find some other to improve English writing.

2. Please confirm whether the spelling and usage of all Latin names are correct before final publication.

3. 3. Please improve the quality of phylogenetic trees because they are in low resolution.

Author Response

Dear reviewer,

Thank you for your valuable suggestions on this article. I have made the following modifications and discussed them below, as per your request.

Please let me know if you have any further suggestions or require additional information.

Thank you for your time and consideration.

Best regards,

Mr. Qianzhen Wu

Reviewer 2 Report

In this study authors list and describe rust fungi on medicinal plants from China. The authors describe three new species and new host parasite relations. The descriptions are ok (not considering the language) including molecular (ITS, LSU) and morphological data.The microscopical and macroscopical illustrations are ok. However, the proposal of three new species is not convincing, also some identifications (see below) doubtful. Except from the taxonomic part it is a nice list of species that is worth to be published. Unfortunately this manuscript suffers from poor language. In case you resubmit it please check the species descriptions. You have to compare them with all similar species, not only with sequences in GenBank. After revising the ms. and before submitting it again I strongly recommend authors to ask an English native speaker AND mycologist  to check the language and revise the manuscript.

Comments on the description of some rust fungi:

Coleosporium senecionis on Aster from China: Sampling poor. For sure this is not Coleosporium senecionis. This species is decribed from Europe and you need to add sequences of Co. senecionis from Senecio (if possible from the type host, you will find sequences in GenBank). This will show you that all sequences from China identified here as Co. senecionis are no Co. senecionis at all but probably a so far undecribed species. So please change the sampling and do not call the fungus Co. senecionis. Also, I don't understand why authors call this fungus Coleosporium asterum in the summery.

Nyssopsora altissima: The closest relative in your phylogeny is N. echinata. I cannot find this sequence in your table of listed species. I do not doubt that there are differences between the two species but I cast some doubt that the species is different from another Nyssopsora on Ailanthus, N. cedrelae. Unfortunately, you do not mention this species and cannot provide a sequence. Consequently, I cannot accept this as a new species. 

Phragmidium cymosum (in Fig. 4 Phragmidium cymosa): No telia are described. I strongly recommend authors to collect specimens with telia before describing a new species that is finally based on molecular data only. Please consider that, that by far, not all Phragmidium species in this complex were sequenced so far. So you have to find the morphological evidence which is only possible with  teliospores in Phragmidium. If you cannot get specimens with telia for this study call it just Phragmidium sp. and do not describe a new species.

Hamaspora alceaefolius: For this species the problem is same as in the previous "new" species. You cannot describe a new species because it is phylogenically different from the only other species of the genus that has been sequenced. There are other Hamaspora spp. on Rubus which have to be sequenced or least significant morphlogical different have to be shown. Other species on Rubus are e.g. H. longissima and H. rubi-sieboldii. 

I am not an English native speaker but my English is good enough to that your English has to be revised (especially arrangement and shortage of sentences). 

Author Response

(The authors gave the same response as above.)

Reviewer 3 Report

The entitled with the manuscript "Rust fungi on medicinal plants in Guizhou Province with descriptions of three new species" was identified rust species. In the research, these species are characterized morphologically and molecular level. More specific details with the manuscript are marked below

In Abstract section

The abstract of the study mentioned that among the described rusts, there were three new species (Hamaspora rubi-alceifolii sp. nov., Nyssopsora altissima sp. nov., and Phragmidium cymosum sp. nov.) and seven known species (Coleosporium asterum, Melampsora laricis-populina, Melampsoridium carpini, Neophysopella ampelopsidis, Nyssopsora koelreuteriae, Phragmidium rosae-roxburghii, P. Tormen tillae). However, upon evaluation of the article, it was indicated that more significant findings were obtained regarding some rust species. For instance, Hamaspora rubi-alceifolii, Nyssopsora altissima, and Phragmidium cymosum were presented as examples of new species. Neophysopella ampelopsidis and Phragmidium tormentillae were first introduced on the Chinese mainland. Melampsora laricis-populina, Melampsoridium carpini, and Nyssopsora koelreuteriae were documented for the first time in Guizhou province. Additionally, Coleosporium senecionis was first found on a new host plant, Aster ageratoides. I believe that these details should be appropriately included in the abstract to better explain the content of the article. The current abstract does not provide sufficient explanation; therefore, it needs to be revised.

In Introduction section

This part of the article provides brief information about rust diseases, but the importance of rust diseases in medicinal plants has not been adequately emphasized. Is there any research on this topic, or if there is no research, what could be the reasons? Additionally, are there any significant rust species identified in medicinal plants in this context? It would be more appropriate to provide information about these aspects that I mentioned.

In Materials and Methods section

It should be clearly stated how many spores, for example mg, were used to isolate DNA from rust spores. In addition, please provide information about the hosts and habitats information of specimens.

The line 109-111 please rewrite as "The Bootstrap values of  ML analyses were obtained by running 1,000 replicates by using a Markov chain Monte Carlo (MCMC) method to approximate the posterior probabilities of trees"

Including both the information of isolates obtained from the current study and retrieved from the genebank should be given as suplementary materials please remove Table 1 (GenBank accession number and information of taxa used for phylogenetic analyses)  in the article.

I think that the other subheading 2.2, 2.3 and 2.4 in the M and M section is sufficent.

Results section

l It is not correct to start the "Results" section directly with a figure. First, the obtained findings should be mentioned, and then the section with figures should follow.

Line 174-176 please give the species name identified in first time.

Line 177. In taxonomy part the evaluations made in this article are presented in appropriate formats

Taxonomy part is suffient for the meetings

Discussion section

I think that the current rust diseases have not been adequately evaluated in this section please revise this section.

 Minor editing of English language required.

Author Response

(The authors gave the same response as above.)

Round 2

Reviewer 2 Report

Phragmidium cymosum sp. nov. The first step you have to go is to check in an additonal field study whether this species forms telia and not to refer to anamorphic Phragmidium spp. If you can confirm that the species does not form telia, everything is fine, if not I recommend to name it Phragmidium sp. Don't you have the possibility just do go there - now (it's a good time to check this at the end of August!). Describing new species and other taxa is involved with major responsibilities, please keep this in mind.

Hamaspora: Please provide an identification key for worldwide Hamaspora spp. on Rubus.

Author Response

Dear reviewer,

Thank you for your valuable suggestions on this article. As a researcher studying rust fungi, I appreciate your advice on maintaining a rigorous approach to identification and taking responsibility for the identification of new species. Thank you for your suggestions, and I will continue to uphold a responsible attitude towards rust fungi assessment in the future.

Please let me know if you have any further suggestions or require additional information.

Thank you for your time and consideration.

Best regards,

Mr. Qianzhen Wu

Reviewer 3 Report

I thank the writers for accepting the suggestions made to them and making the necessary corrections. I wish them success in their future studies

Author Response

Dear reviewer,

Thank you for your valuable feedback on this article. I sincerely wish you continuous academic growth and happiness in your family.

Best regards,

Mr. Qianzhen Wu